# Asynchronous digital health interventions for reviewing asthma: A mixed-methods systematic review protocol

**Md. Nazim Uzzaman**[1], **Vicky Hammersley**[1], **Kirstie McClatchey**[1], **Jessica Sheringham**[2], **G. M. Monsur Habib**[1,3], **Hilary Pinnock**[1]*

**1** Asthma UK Centre for Applied Research, Usher Institute, The University of Edinburgh, Edinburgh, Scotland, United Kingdom, **2** Institute of Epidemiology & Health, University College London, London, England, United Kingdom, **3** Community Respiratory Centre, Bangladesh Primary Care Respiratory Society, Khulna, Bangladesh

* hilary.pinnock@ed.ac.uk

**Data Availability Statement:** No datasets were generated or analysed for this systematic review protocol. Summary tables and syntheses from the

## Abstract

### Introduction

People living with asthma require regular reviews to address their concerns and questions, assess control, review medication, and support self-management. However, practical barriers to attending face-to-face consultations might limit routine reviews. Reviewing asthma using asynchronous digital health interventions could be convenient for patients and an efficient way of maintaining communication between patients and healthcare professionals and improving health outcomes. We, therefore, aim to conduct a mixed-methods systematic review to assess the effectiveness of reviewing asthma by asynchronous digital health interventions and explore the views of patients and healthcare professionals about the role of such interventions in delivering asthma care.

### Methods

We will search MEDLINE, Embase, Scopus, PsycInfo, CINAHL, and Cochrane Library from 2001 to present without imposing any language restrictions. We are interested in studies of asynchronous digital health interventions used either as a single intervention or contributing to mixed modes of review. Two review authors will independently screen titles and abstracts, and retrieve potentially relevant studies for full assessment against the eligibility criteria and extract data. Disagreements will be resolved by discussion with the review team. We will use 'Downs and Black' checklist, 'Critical Appraisal Skills Programme', and 'Mixed Methods Appraisal Tool' to assess methodological quality of quantitative, qualitative, and mixed-methods studies respectively. After synthesising quantitative (narrative synthesis) and qualitative (thematic synthesis) data separately, we will integrate them following methods outlined in the Cochrane Handbook for Systematic Reviews of Interventions.

### Conclusion

The findings of this review will provide insights into the role of asynchronous digital health interventions in the routine care of people living with asthma.

review will be made publicly available in a peer-reviewed publication.

**Funding:** MNU is supported by a University of Edinburgh College of Medicine PhD Studentship (Grant number 34678) funded by the University of Edinburgh College of Medicine and Veterinary Medicine (CMVM) within the Asthma UK Centre for Applied Research (AUKCAR). The PhD studentship is nested in the IMP2ART (IMPlementing IMProved Asthma self-management as RouTine) programme at the University of Edinburgh (https://www.ed.ac.uk/usher/imp2art). The funders had no role in study design, data collection and analysis, decision to publish, or preparation of the manuscript.

**Competing interests:** The authors have declared that no competing interests exist.

## Trial registration

**Systematic review registration:** PROSPERO registration number: CRD42022344224.

## Introduction

Asthma is a common, long-term airway disease affecting up to 18% of all age groups globally [1, 2]. Although, hospitalisation and deaths from asthma have declined in some countries, asthma continues to exert an unacceptably high burden on healthcare systems and society, resulting in reduced productivity at work and social disruption [2]. National and international guidelines recommend that people with asthma should be provided with self-management education reinforced by a personalised asthma action plan and supported by regular review to improve their control over their asthma [3–5]. An asthma review is a routine check-up of people with asthma to assess control, respond to that assessment by adjusting the management strategy, as well as to explore patients' thoughts, concerns, and expectations, and to guide self-management [6, 7]. Asthma reviews should be completed regularly (at least annually in stable patients) as a scheduled appointment [8]. A more frequent review may be necessary when a diagnosis is first made or for those who have poor asthma control [8]. However, across the United Kingdom (UK) National Health Service (NHS), 1 in 20 patients miss general practice (GP) appointments [9], and asthma clinics have higher than average proportion of missed appointments [10]. Practical barriers such as geographical distance, work commitments, transportation time and cost, long waiting time to attend face-to-face consultations may be barriers to regular reviews of asthma [11].

Digital health uses innovative information and communication technology to meet health demands. The term 'digital health' is an umbrella term encompassing eHealth, mHealth, health information technology, wearable devices, telehealth, telemedicine, and increasingly is linked with advanced computing such as machine learning and artificial intelligence [12, 13]. The World Health Organization (WHO) in 2005 urged member states to draw up a strategic plan for promotion of equitable, affordable and universal access to the benefits of digital health services [14]. In 2012, the European health policy framework-Health 2020 highlighted the importance of digital technology in advancing public health priorities and achieving the health-related Sustainable Development Goals [15, 16]. In the UK [17], there has been a drive over the last decade to increase the use of digital health interventions for managing and monitoring people with long-term conditions at home, and reduce the need for avoidable visits to their GP practice and hospital. For asthma, a condition that places a substantial burden on healthcare systems, using digital health interventions to review asthma could be convenient for patients and an efficient way of maintaining communication between patients and healthcare professionals and improving health outcomes [18].

Digital health interactions are typically categorised as: synchronous or real-time, a live consultation (for example, videoconferencing between patients and healthcare professionals); or asynchronous or 'store-and-forward', a non-concurrent consultation (for example, transmission of clinical data from patients through email that allows a healthcare professional to review the data and provide feedback at a later time) [19, 20]. Synchronous remote asthma reviews using telephone or videoconferencing have become mainstream during the COVID-19 pandemic [21, 22]. They are relatively well investigated and have been shown to increase asthma review rates without clinical disadvantage or loss of satisfaction [23–27]. Asynchronous reviews may overcome the temporal limitations of in-person and remote synchronous care, and have the potential to support the care of large numbers of patients with asthma [28]. Existing systematic reviews (see Table 1) have synthesised the evidence for a broad range of digital

**Table 1. Summary of relevant published reviews (ordered by publication year).**

| Author-Year | Objective | Main outcomes | Key conclusions | Recommendations by the author | Outstanding gaps |
|---|---|---|---|---|---|
| McLean 2010 [34] | To assess the effectiveness of telehealthcare interventions in people with asthma | • Quality of life | • Telehealthcare initiatives are unlikely to improve quality of life for most people with mild asthma | Future reviews should include more networked and internet-based interventions | • Synchronous and asynchronous telehealthcare interventions were not distinguished |
| | | • Emergency department visit | • May be useful in preventing exacerbations and hospital admissions | | • Views and experiences of patient and/or HCPs were not explored |
| | | • Hospitalisation | | | • Process outcomes were not measured |
| Morrison 2014 [35] | To summarise the effectiveness and implementation of digital asthma self-management | • Activity limitation | • Digital self-management interventions are promising and have been shown to improve some outcomes | Further systematic review to know the currently available digital interventions and examine stakeholders' perspectives | • This is a meta-review |
| | | • Adverse events | | | • Qualitative component needs further exploration |
| | | • Barriers/ facilitators | | | |
| | | • Health service utilisation | | | |
| | | • Medication use | | | |
| | | • Quality of life | | | |
| | | • Asthma control | | | |
| McLean 2016 [33] | To summarise interactive digital interventions to support asthma self-management and determine impact | • Clinical outcomes | • Digital self-management interventions for adults with asthma show promise, with small beneficial effects on asthma control | Further study to assess the effect of interactive digital interventions on mental health | • Did not include child and adolescent populations |
| | | • Cost effectiveness | | | • Views and experiences of patient and/or HCPs were not explored |
| Kew 2016 [18] | To assess the safety and efficacy of conducting asthma check-ups remotely compared to usual face-to-face consultations | • Exacerbations | • There was no difference in asthma control and quality of life between remote and face-to-face check-ups | Further studies to include remote monitoring and remote check-ups in the interventions | • Synchronous and asynchronous digital health interventions were not distinguished |
| | | • Asthma control | | | • Views and experiences of patient and/or HCPs were not explored |
| | | • Serious adverse events | | | |
| Kew 2016 [32] | To assess the efficacy and safety of home telemonitoring with HCP feedback between clinic visits | • Exacerbations | • Unsure whether additional telemonitoring strategies improve symptom control or reduce need for oral steroids over usual care | Qualitative studies could inform future research by focusing on patient and provider preferences | • Process outcomes were not measured |
| | | • Asthma control | | | |
| | | • Serious adverse events | | | |
| Chongmelaxme 2018 [30] | To determine the effects of telemedicine on asthma control and the quality of life in adults | • Asthma control | • Combined-telemedicine involving tele-case management or tele-consultation were effective for asthma control and improving quality of life | Future research to assess economic, ethical, legal, and sociocultural aspects before implementing various telemedicine interventions | • Views and experiences of patient and/or HCPs were not explored |
| | | • Quality of life | | | • Process outcomes were not measured |
| Jeminiwa 2019 [31] | To assess effectiveness of eHealth in improving adherence to inhaled corticosteroids and explore satisfaction of patients | • Adherence to inhaled corticosteroids | • eHealth interventions are effective and acceptable in improving patient adherence to inhaled corticosteroids | Future studies should employ either objective measures of adherence or validated self-report instruments | • Synchronous and asynchronous digital health interventions were not distinguished |
| | | | | | • Process outcomes were not measured |
| Snoswell 2020 [37] | To examine the change in quality of life after interactive telehealth interventions and explore effective telehealth modalities | • Quality of life | • Interactive telehealth interventions improved quality of life | An updated review of evidence with a clear definition of telehealth is needed | • Synchronous and asynchronous digital health interventions were not distinguished |
| | | | | | • Views and experiences of patient and/or HCPs were not explored |

*(Continued)*

**Table 1.**  (Continued)

| Author-Year | Objective | Main outcomes | Key conclusions | Recommendations by the author | Outstanding gaps |
|---|---|---|---|---|---|
| Mosnaim 2021 [36] | To explore available digital health interventions and assess the future utility of digital health technology in asthma | • Adherence to inhaled corticosteroids<br><br>• Asthma impairment<br><br>• Healthcare use | Interventions featuring non-individualised content improved adherence to inhaled corticosteroids, but with no improvement in asthma burden | Future research into digital technology as a part of asthma management is required | • Scoping review plotting available asthma digital health interventions<br><br>• Did not distinguish between synchronous and asynchronous communication between patients/carers and HCPs |
| Chan 2022 [29] | To assess the effectiveness of digital interventions for improving adherence to maintenance treatments in asthma | • Adherence to maintenance medication<br><br>• Asthma control<br><br>• Exacerbations requiring oral corticosteroids | Digital technologies may help people with asthma to adhere to maintenance treatment, improve asthma control, and quality of life | Further research is needed to identify the components of effective digital adherence interventions | • Synchronous and asynchronous digital health interventions were not distinguished<br><br>• Views and experiences of patient and/or HCPs were not explored |

HCPs = Healthcare professionals.

technologies, telemonitoring and telehealth (the terminology is used inconsistently) [18, 29–37] but there are no reviews synthesising the evidence for the effectiveness specifically of asynchronous digital health interventions for routine asthma care, nor exploring the views and experiences of patients and/or professional stakeholders on their utility. We therefore aimed to systematically review the qualitative and quantitative evidence to derive recommendations for policy and practice on the use of asynchronous digital health interventions for reviewing asthma.

## Review questions

Specifically the review questions are:

1.  How are asynchronous digital health interventions used for reviewing asthma?

1.1. What digital health functionality is used?

1.2. How is digital health incorporated into routine asthma care?

2.  What are the effects of asynchronous digital health interventions on asthma control, acute attacks, quality of life, and other healthcare outcomes compared to usual care or no review consultation?

3.  What are the views and experiences of patients, and/or healthcare professionals on asynchronous digital health interventions for reviewing people with asthma in terms of:

3.1. Acceptability for receiving or providing care for individuals?

3.2. Organisational approaches to delivering care?

4.  From the quantitative and qualitative synthesis, what findings (if any) can be applied to clinical practice and policymaking?

5.  What are the gaps in existing research?

## Methodology

We will follow the methodology in the Cochrane Handbook for Systematic Reviews of Interventions to conduct this mixed-methods review [38]. We will follow a results-based convergent design where qualitative and quantitative data will be analysed and presented separately but integrated using a further synthesis [39]. The review is registered with PROSPERO (ID: CRD42022344224), any changes to the published record will be reported.

### Search strategy

One review author (MNU) will develop a search strategy involving the review team (HP, VH, KM, JS and MH) and a senior librarian from the University of Edinburgh. MNU will identify records through searching the following databases: MEDLINE, Embase, Scopus, PsycInfo, CINAHL, and Cochrane Library (S1 Appendix). We will search the databases from 2001 because access to the internet increased after the introduction of third-generation (3G) cellular technologies, and interactive asynchronous communication thus became a viable option for more people [40]. We will not impose any restriction on language of publication during database searching and arrange translation to English of potentially relevant quantitative studies to enable selection and data extraction [41]. However, we will only consider qualitative and mixed-methods studies written in English because of the loss of nuance with language translation [42] but we will provide a brief description in the final results. We will conduct a pre-publication update by checking the reference lists and conducting forward citation of all studies selected for additional eligible studies [43].

### Study selection

Following the search, all identified citations will be downloaded into EndNote 20 (Clarivate Analytics, PA, USA) and duplicates removed using SRA Deduplicator software [44]. Two authors (MNU and MH) will independently screen titles and abstracts, retrieve and review full-text papers for inclusion of studies against the eligibility criteria (see Table 2) using Covidence (www.covidence.org) [45]. Reasons for exclusion of full-text studies that do not meet the inclusion criteria will be recorded and reported. Any disagreements that arise between the two reviewers (MNU and MH) at any stage of the study selection process will be resolved through discussion and involve the review team (HP, VH, KM, JS) if necessary. The results of the search will be presented in a Preferred Reporting Items for Systematic Reviews and Meta-analyses (PRISMA) flow diagram [46].

### Data extraction and management

Two review authors (MNU and MH) will pilot the data extraction form on at least one quantitative and one qualitative study before data are extracted from the remaining included studies using a refined form. We will extract data into a Microsoft Word and Excel file as necessary. Two review authors (MNU and MH) will independently extract data from all included studies (quantitative, qualitative and mixed-methods) and another author (HP/VH/KM/JS) will check accuracy of data transcribed into tables or meta-analyses. Any disagreement between MNU and MH relating to data extraction will be resolved by consensus. A third review author (HP/VH/KM/JS) will be involved to resolve any outstanding disagreement as necessary.

**Quantitative studies.** Two review authors (MNU and MH) will independently extract the following study characteristics from included studies.

- Participants: number, mean age, gender, severity of condition, diagnostic criteria, baseline lung function, smoking history, inclusion criteria and exclusion criteria.

**Table 2. Inclusion and exclusion criteria, and operational rules.**

| | Description, inclusion | Exclusion criteria | Operational rules |
|---|---|---|---|
| Population | • Children (and their caregivers) and adults with a primary diagnosis of asthma<br><br>• Comorbidity will not be an exclusion criterion as long as the focus of the intervention is asthma | Studies that recruited participants with other long-term conditions, unless they report data for people with asthma. | |
| Intervention | • Reviewing asthma by asynchronous digital health interventions | • Exclusively synchronous or real-time review of asthma by any means such as face-to-face consultations, video-conferences, telephone calls etc. | **'Reviewing asthma by asynchronous digital health interventions'**- the key criteria are that:<br><br>1. Exchange of relevant information or notes between patients/carers and their HCPs (e.g., any symptoms, triggers, concerns or questions, lung function measurements, medications, action plan) and/or share necessary documents (e.g, images, videos of inhaler technique, asthma control measures) as part of reviewing asthma and decision making. AND |
| | • Concomitant face-to-face or synchronous reviews will not be an exclusion criterion as long as a proportion of the care is provided by asynchronous digital interventions | • Acute asthma consultations | 2. Use of any forms of digital health interventions including telehealth, telemedicine, mHealth, eHealth, health information technology, and 'Internet of things' (IoT) for delivery of the intervention. AND |
| | | | 3. 'Store and forward' or asynchronous or non-concurrent communications between patients/carers and their HCPs |
| Comparison (Quantitative study) | • Either population receiving 'Usual care' OR | | • **'Usual Care'**- is the standard face-to-face asthma review received by an individual with asthma in the any healthcare system |
| | • receiving care exclusively by 'synchronous remote reviews' | | • **'Exclusively synchronous remote reviews'**- are the real-time or concurrent reviewing of asthma by any mode of consultation. |
| | • OR | | |
| | • no review consultation | | |
| Outcomes (Quantitative study) | One or more of the following outcomes: | | Clinical outcome measurement: |
| | **Clinical outcomes** | | Priority will be given to validated instruments for measuring asthma control (e.g. Asthma Control Test [47]) or quality of life (e.g. mini Asthma Quality of Life Questionnaire [48]), and acute attacks as defined by ATS/ERS Task force [49] and measured by unscheduled healthcare visits/admissions requiring a steroid course. |
| | Asthma control | | |
| | Acute attack | | |
| | Asthma-related quality of life | | |
| | **Process outcomes** | | |
| | Health service time: Time to first response; duration of consultation(s); time spent resolving technical issues | | |
| | Patient time: Time completing consultation tasks | | |
| | Conversion to synchronous consultation | | |
| | Uptake, ease of access to care, access to information, interactivity | | |
| | **Others** | | |
| | Outcomes suggested by qualitative studies, user satisfaction, self-care, environmental impact, adverse events | | |
| Phenomena of interest (Qualitative study) | Studies that explored views and experiences of patients, and/or professional stakeholders on asynchronous digital health interventions (with or without other modes of communication) for reviewing asthma | Studies that did not include any views or opinions about asynchronous consultations | Data about the views and experiences of patients, and/or professional stakeholders on asynchronous digital interventions (either as an isolated intervention or as an adjunct to other forms of consultation) for reviewing asthma will be extracted for the synthesis of findings |
| Setting and context | Any countries and healthcare settings irrespective of economic status | | |

*(Continued)*

**Table 2.** (Continued)

| | Description, inclusion | Exclusion criteria | Operational rules |
|---|---|---|---|
| Study designs | Quantitative: RCTs; CCTs, observational studies, pre-post studies | Case study, case report, editorials, letter to editor, commentary, reviews, expert opinion articles, and conference abstracts. | Conference abstracts will be excluded, but will prompt a search for a subsequent published paper. |
| | Qualitative studies (including observational studies; content analysis, documentary analysis) | | |
| | Mixed-methods studies | | |
| Language | • Quantitative studies: no language restriction | Qualitative studies and qualitative component of mixed-methods studies published in languages other than English | The search will not be restricted by language. Translation to English will be arranged for quantitative studies and the quantitative component of mixed-methods studies to enable selection and data extraction. Translation to English will not be arranged for qualitative studies and qualitative component of mixed-methods studies (because of the difficulty of reflecting nuanced qualitative data). |
| | • Mixed-methods studies: no language restriction for quantitative component | | |
| | • Qualitative studies: English only | | |

HCP = Healthcare professional; RCT = randomised controlled trial; CCT = controlled clinical trial.

- Methods: study design, duration of study, number of study centres and location, study setting, and date of study.

- Interventions: intervention, comparison, concomitant medications and excluded medications.

- Outcomes: primary and secondary outcomes specified and collected: e.g., mean (standard deviation), median (interquartile range), confidence intervals, $P$-values, measurement scales used, and time points reported.

- Notes: funding for studies and conflicts of interest of trial authors.

**Qualitative studies.** Two review authors (MNU and MH) will extract the following study characteristics from included studies.

- Study details: country, study type (e.g. focus group, semi-structured interviews, structured interviews), dates, source of funding, objectives.

- Participants: number, mean age, gender, severity of condition, diagnostic criteria, baseline lung function, smoking history, inclusion criteria and exclusion criteria.

- Methods: sampling, setting (e.g. community or outpatient or hospital), data collection (e.g. how the authors conducted the study, length of interviews, whether interviews were recorded, use of interview guide), data analysis (e.g. method of analysis of transcripts, framework used, coding, thematic map).

- Results: themes and quotes from participants, and authors' interpretations.

**Mixed-method studies.** Two review authors (MNU and MH) will independently extract the study characteristics as listed above for the quantitative and qualitative components separately from included studies.

## Dealing with missing data

We will use email to contact study authors for any unreported data or clarification of study methodologies. If data is still unavailable, we will analyse the available data and reflect on the significance of missing data in the discussion section.

## Methodological quality assessment

All included studies (quantitative, qualitative and mixed-methods) will be assessed for methodological quality independently and in duplicate by two review authors (MNU and MH). We will resolve any disagreements by discussion or by involving another author (HP/VH/KM/JS) if necessary.

**Quantitative studies.** To assess the methodological quality of randomised controlled trials (RCTs), we will use the Cochrane risk of bias tool which assesses selection, performance, detection, attrition, reporting and other sources of bias enabling each study to be assigned as low; moderate; or high risk of bias [50]. We will record and tabulate a summary of the assessment with the overall judgement. For non-randomised studies, we will use the Downs and Black checklist [51]. To reflect the relative weight of the quantitative findings, we will adopt the previously published approach of summarising three attributes for each study (design, population size, quality score) when presenting data from the different studies [52].

**Qualitative studies.** We will assess study quality by identifying methodological strengths and limitations (i.e., rigour) of included studies using the Critical Appraisal Skills Programme (CASP) quality assessment tool for qualitative studies [53], following the domains recommended by the Cochrane Qualitative and Implementation Methods Group [54]:

- Clarity of aims and research question

- Congruence between the research question and design

- Rigour of case and/or participant identification, sampling, and data collection to address the question

- Proper application of the method; conceptual depth of findings, exploration of deviant cases and alternative clarifications, and reflexivity of the researchers

- We will present the quality assessment findings in a table.

**Mixed-methods studies.** We will use the Mixed Methods Appraisal Tool (MMAT) to assess risk of bias [55]. We will assess the risk of bias according to the following criteria:

- Is there an adequate rationale for using a mixed-methods design to address the research question?

- Are the different components of the study effectively integrated to answer the research question?

- Are the outputs of the integration of qualitative and quantitative components adequately addressed?

- Are divergences and inconsistencies between quantitative and qualitative results adequately addressed?

- Do the different components of the study adhere to the quality criteria of each tradition of the methods involved?

## Data synthesis

**Quantitative data.** Based on our initial scoping we anticipate that our included studies will have substantial clinical, methodological and statistical heterogeneity and meta-analysis may not be appropriate. If that is the case, we will do a narrative synthesis to show the major outcomes and their relationships [50], illustrating findings graphically if appropriate [56].

However, if we find sufficient number of RCTs, we will perform meta-analysis for the clinical outcomes (asthma control, acute attacks, and asthma-related quality of life). One review author (MNU) will conduct the meta-analysis using Review Manager software (RevMan 2020, V.5.4.1) and another review author (MH) will check data accuracy. We will conduct a pooled quantitative synthesis for homogeneous data from RCTs using an inverse variance method and a random-effects model in the meta-analysis. If the included RCTs use the same outcome measurement tool, we will use pooled mean differences. However, if (as expected) outcome measurement tool varies among trials, we will consider standardised mean differences.

**Qualitative data.** We will use thematic synthesis to combine the findings of studies that describe the views and experiences of patients and healthcare professionals on asynchronous asthma reviews. Following recognised methodology [57], two review authors (MNU and MH) will begin by familiarising themselves with the data against the aims of the review and note recurrent themes across the studies. After that they will develop a coding framework in discussion with the review team (HP, VH, KM, and JS). MNU and MH will independently perform line-by-line initial coding of the findings of the included studies (defined as all the text/quotes under the heading of 'results' or 'findings') translating the concepts from one study to another. They will then search for themes according to the predetermined thematic framework adding additional themes as they emerge. Analysis will be iterative and involve the multi-disciplinary author team before finalising the overarching themes and sub-themes. We will initially analyse patients and healthcare professionals' data separately to identify, for example, conflicting views or experiences. If we find that the views and experiences are similar, we may combine the two subgroups in subsequent syntheses. We will generate tables of author-reported categories, themes, and subthemes regarding asynchronous online asthma reviews.

**Combining quantitative and qualitative data.** After synthesising quantitative and qualitative data separately, we will integrate them following the methods and recommendations outlined in the Cochrane Handbook [58]. We will choose the appropriate methods and tools for integration as the review progresses following the Cochrane Qualitative and Implementation Methods Group guidance [59]. We anticipate that we will juxtapose the quantitative and qualitative findings in a matrix. The findings from the qualitative evidence synthesis (e.g. intervention components linked to acceptability or feasibility of the interventions) will drive juxtaposition, and these findings will make up one side of the matrix. The other side of the matrix will contain findings on intervention effects (e.g., improves outcome, no difference in result, unknown impacts). The presence or absence of features indicated by the hypotheses obtained from the qualitative synthesis will be used to categorise quantitative studies based on findings on intervention effects and the presence or absence of features specified by the qualitative synthesis [60]. Observed patterns in the matrix (if any) will be used to explain variations in quantitative study findings and to identify research gaps [61].

Interpretation will be aided by discussion within the multidisciplinary team and with the insights of patient and public involvement colleagues from the Asthma UK Centre for Applied Research (https://www.ed.ac.uk/usher/aukcar).

## Assessment of confidence in evidence

**Quantitative data.** We will use the five GRADE (Grading of Recommendations Assessment, Development and Evaluation) considerations (risk of bias, consistency of effect, imprecision, indirectness, and publication bias) to assess the quality of evidence for the primary quantitative outcomes following the methods and recommendations described in the Cochrane Handbook [62]. We will use GRADEpro GDT software [63] and provide footnotes to explain any decisions to downgrade the quality of evidence.

**Qualitative data.** We will follow the methods and recommendations of the Cochrane Qualitative and Implementation Methods Group [54] and use the Grades of Recommendation, Assessment, Development, and Evaluation–Confidence in the Evidence from Qualitative Reviews (CERQual) approach to assess confidence in synthesised qualitative findings [64]. CERQual includes four domains: methodological limitations, relevance of contributing studies to the research question, coherence of study findings, and adequacy of data supporting the study findings. We will summarise findings of the four domains for each outcome and provide justification to explain any decisions to downgrade the quality of evidence.

## Dissemination

In addition to publishing in a peer-reviewed journal, we will share our review findings at national and international scientific meetings and conferences. Additionally, we will employ innovative dissemination techniques including online seminars and social media.

## Discussion

Remote consultation with limited face-to-face contact is likely to become an important component of global models of asthma care. Reviewing asthma by asynchronous digital health interventions has the potential to prompt timely intervention and improve several areas of asthma management such as disease disparity, medication adherence, patient-clinician communication, supported self-management and make future asthma management more proactive. Asynchronous digital interventions for reviewing asthma are likely to be convenient, but little is known about how this technology is being used, if/how it is acceptable and useful to patients, and if it is perceived as effective and safe by the professionals in different healthcare settings across the world. The findings of this review are expected to provide valuable insight into organising routine care for people living with asthma in the context of multiple modes of consulting in a way that benefits both patients and healthcare professionals.

## Supporting information

**S1 Checklist. PRISMA-P 2015 checklist.**
(DOCX)

**S1 Appendix. Database search strategy.**
(DOCX)

## Acknowledgments

We acknowledge the contribution of Marshall Dozier, College Lead for Library Academic Support at the University of Edinburgh, and Bohee Lee, Asthma UK Centre for Applied Research PhD student in helping develop the search strategy.

## Author Contributions

**Conceptualization:** Md. Nazim Uzzaman, Vicky Hammersley, Kirstie McClatchey, Jessica Sheringham, Hilary Pinnock.

**Data curation:** Md. Nazim Uzzaman, G. M. Monsur Habib.

**Funding acquisition:** Vicky Hammersley, Hilary Pinnock.

**Investigation:** Md. Nazim Uzzaman, Vicky Hammersley, Hilary Pinnock.

**Methodology:** Md. Nazim Uzzaman, Kirstie McClatchey, Jessica Sheringham, G. M. Monsur Habib, Hilary Pinnock.

**Project administration:** Vicky Hammersley, Hilary Pinnock.

**Supervision:** Vicky Hammersley, Kirstie McClatchey, Jessica Sheringham, Hilary Pinnock.

**Visualization:** Md. Nazim Uzzaman.

**Writing – original draft:** Md. Nazim Uzzaman.

**Writing – review & editing:** Vicky Hammersley, Kirstie McClatchey, Jessica Sheringham, G. M. Monsur Habib, Hilary Pinnock.

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
