## [Decision Letter · Decision Letter 0]

17 Nov 2022

PONE-D-22-22457Asynchronous digital health interventions for reviewing asthma: a mixed-methods systematic review protocolPLOS ONE

Dear Dr. Pinnock,

Thank you for submitting your manuscript to PLOS ONE. After careful consideration, we feel that it has merit but does not fully meet PLOS ONE’s publication criteria as it currently stands. Therefore, we invite you to submit a revised version of the manuscript that addresses the points raised during the review process.Please find review comments at the bottom of this email. I note that you will use a 3rd reviewer to resolve any conflicts for the quantitative data extraction but not the qualitative or mixed methods data extraction. Is there a reason for this or is it an error? As highlighted by reviewer 1 please provide more information on how you plan to undertake the thematic synthesis. 

We look forward to receiving your revised manuscript.

Kind regards,

Heather Leggett

Academic Editor

PLOS ONE

Journal Requirements:

We note that the grant information you provided in the ‘Funding Information’ and ‘Financial Disclosure’ sections do not match.

“The authors received no specific funding for this work. This paper is part of a PhD programme of research.”

Reviewers' comments:

Reviewer's Responses to Questions

**Comments to the Author**

1. Does the manuscript provide a valid rationale for the proposed study, with clearly identified and justified research questions?

Reviewer #1: Yes

Reviewer #2: Yes

2. Is the protocol technically sound and planned in a manner that will lead to a meaningful outcome and allow testing the stated hypotheses?

Reviewer #1: Yes

Reviewer #2: Yes

3. Is the methodology feasible and described in sufficient detail to allow the work to be replicable?

Reviewer #1: Yes

Reviewer #2: Yes

4. Have the authors described where all data underlying the findings will be made available when the study is complete?

Reviewer #1: No

Reviewer #2: Yes

5. Is the manuscript presented in an intelligible fashion and written in standard English?

Reviewer #1: Yes

Reviewer #2: Yes

6. Review Comments to the Author

You may also provide optional suggestions and comments to authors that they might find helpful in planning their study.

Reviewer #1: Overall, this is a comprehensive protocol with several standout points of note including: the inclusion of non-English language papers, matrix to synthesise mixed methods data, and inclusion of PPI members.

There are several points of note I would like the authors to respond to:

Line 86: The authors have included in brackets that Asthma reviews should be completed “at least annually in stable patients” without providing a reference. Please provide a reference.

Line 102: The European Health Policy Framework – Health 2020 is cited but the references in this sentence don’t appear to match the online report. Please double check reference 15/16 covers this report.

Line 147: Review question 4 is poorly worded and difficult to understand, recommend a rewrite.

Line 162: How are you/have you developed the search strategy? I would like details on how it was formulated and who was involved.

Line 168: You aren’t planning on translating and including non-English language papers for qualitative papers due to ‘loss of nuance with language translation’. You don’t provide any references and evidence to support this statement and I’m not sure without it I would agree that’s enough of a reason. Please provide evidence to support that this statement.

Line 181: It’s not actually stated who and how many people are conducting full-text review.

Line 184 - 186: Doesn’t make sense, recommend a reword.

Line 192: What is the reasoning for only having one author pilot the extraction form when two authors will be independently using it to extract the same data? I would advise having both authors pilot the form to make sure the same information is being extracted and ensure consistency.

Line 196: It states that two authors are conducting data extraction. It’s not clear if both authors are extracting all data or if they’re doing a proportion each that are then being checked by a third author, please clarify.

Line 206: Are you collecting raw data (e.g.: means, standard deviation, mean difference). At the moment it’s unclear if you’re collecting summary data only.

Line 256: Risk of bias is discussed quite minimally, with no discussion of who is conducting it, how disagreements will be resolved, and how the information will be presented. Ideally there should be more information here.

Line 270:. It’s stated that it’s not anticipated a meta-analysis will be possible, but as you don’t know that yet what is your plan if a meta-analysis is possible?

Line 276: It is stated that thematic synthesis will be used without any information of what that practically means. Are you using line-by-line, who will be coding, are you going so far as to produce analytical themes or purely descriptive ones?

Line 288: Double check your citations meet the PLOS ONE guidelines, the ‘Harden and colleagues [60]’ does not look correct.

In line with PLOS ONE’s data policy, please describe how the data will be made available at the point of publication.

Reviewer #2: Thank you for this clearly written protocol with all necessary questions addressed. Helpful, but not needed, to identify who will be doing what at each stage, as this is only done for some tasks (eg. screening).

7. PLOS authors have the option to publish the peer review history of their article (what does this mean?). If published, this will include your full peer review and any attached files.

Reviewer #1: No

Reviewer #2: **Yes: **Amanda Ross-White

---

## [Author Response · Author response to Decision Letter 0]

10 Dec 2022

Academic Editor: Dr Heather Leggett

1. Thank you for submitting your manuscript to PLOS ONE. After careful consideration, we feel that it has merit but does not fully meet PLOS ONE’s publication criteria as it currently stands. Therefore, we invite you to submit a revised version of the manuscript that addresses the points raised during the review process.

Thank you for your encouraging words and for enclosing the helpful comments from your reviewers. We have responded to the reviewers’ comments in details below. 

2. I note that you will use a 3rd reviewer to resolve any conflicts for the quantitative data extraction but not the qualitative or mixed methods data extraction. Is there a reason for this or is it an error? As highlighted by reviewer 1 please provide more information on how you plan to undertake the thematic synthesis. 

Thank you for highlighting the need for further clarity about data extraction. In fact, we will involve a 3rd reviewer to resolve any conflicts relating to data extraction for all types of included studies. We intended to avoid repetition of the same information in the consecutive subsections. However, the information should have been placed under the ‘Data extraction and management’ section instead of ‘Qualitative studies’ subsection. We now have updated the paragraph under the ‘Data extraction and management’ section and the text reads: 

“Two review authors (MNU and MH) will pilot the data extraction form on at least one quantitative and one qualitative study before data are extracted from the remaining included studies using a refined form. We will extract data into a Microsoft Word and Excel file as necessary. Two review authors (MNU and MH) will independently extract data from all included studies (quantitative, qualitative and mixed-methods) and another author (HP/VH/KM/JS) will check accuracy of data transcribed into tables or meta-analyses. Any disagreement between MNU and MH relating to data extraction 

 will be resolved by consensus. A third review author (HP/VH/KM/JS) will be involved to resolve any outstanding disagreement as necessary.” [Page 15, line 197-206]

Also, we now have removed the similar text from the ‘Quantitative studies’ subsection. Please see page 15, line 209-211. 

Regarding the comments about thematic synthesis, we have detailed our response to the reviewer’s comment #Q14. For your convenience, we reproduce our response below:

“We will use thematic synthesis to combine the findings of studies that describe the views and experiences of patients and healthcare professionals on asynchronous asthma reviews. Following recognised methodology [58], two review authors (MNU and MH) will begin by familiarising themselves with the data against the aims of the review and note recurrent themes across the studies. After that they will develop a coding framework in discussion with the review team (HP, VH, KM, and JS). MNU and MH will independently perform line-by-line initial coding of the findings of the included studies (defined as all the text/quotes under the heading of ‘results’ or ‘findings’) translating the concepts from one study to another. They will then search for themes according to the predetermined thematic framework adding additional themes as they emerge. Analysis will be iterative and involve the multi-disciplinary author team before finalising the overarching themes and sub-themes. We will initially analyse patients and healthcare professionals’ data separately to identify, for example, conflicting views or experiences. If we find that the views and experiences are similar, we may combine the two subgroups in subsequent syntheses. We will generate tables of author-reported categories, themes, and subthemes regarding asynchronous online asthma reviews.” [Page 19-20, line 304 - 322]

Reviewer #1: 

Q1. Overall, this is a comprehensive protocol with several standout points of note including: the inclusion of non-English language papers, matrix to synthesise mixed methods data, and inclusion of PPI members. There are several points of note I would like the authors to respond to:

Thank you for your encouraging comment. We appreciate your time to review and help us improve the manuscript. 

Q2. Line 86: The authors have included in brackets that Asthma reviews should be completed “at least annually in stable patients” without providing a reference. Please provide a reference.

Thanks for bringing this to our attention. We now have provided a reference to support the statement as you suggest. Please see page 4, line 87 and the reference #8.

Reference 8: BTS/SIGN Guideline for the management of asthma 2019. https://www.brit-thoracic.org.uk/quality-improvement/guidelines/asthma/. Last accessed: 2nd December 2022. [Page 24, line 433-435]

Q3. Line 102: The European Health Policy Framework – Health 2020 is cited but the references in this sentence don’t appear to match the online report. Please double check reference 15/16 covers this report.

Thank you for spotting this error. We now have replaced the reference number 15 with the correct one. The reference number 16 is correct.

Reference 15: Health 2020: A European policy framework and strategy for the 21st century. https://apps.who.int/iris/bitstream/handle/10665/326386/9789289002790-eng.pdf?sequence=1&isAllowed=y. Last accessed: 2nd December 2022. [Page 24, line 457-460]

Q4. Line 147: Review question 4 is poorly worded and difficult to understand, recommend a rewrite.

We have revised the review question 4 as you suggest. The text now reads:

“From the quantitative and qualitative synthesis, what findings (if any) can be applied to clinical practice and policymaking?” [Page 9, line 147-152]

Q5. Line 162: How are you/have you developed the search strategy? I would like details on how it was formulated and who was involved.

The search strategy has been developed by the 1st author (MNU) who is a PhD student. Initially, he drafted the search terms and discussed them with his supervisors (HP, VH, KM, and JS). He then developed the syntaxes and finalised the search strategy with the help of a senior librarian at the University of Edinburgh. We have acknowledged the librarian support in our manuscript. We now have updated the sentence in the ‘Search strategy’ which reads: 

“One review author (MNU) will develop a search strategy involving the review team (HP, HV, KM, JS and MH) and a senior librarian from the University of Edinburgh.” [Page 10, lines 164-168]

Q6. Line 168: You aren’t planning on translating and including non-English language papers for qualitative papers due to ‘loss of nuance with language translation’. You don’t provide any references and evidence to support this statement and I’m not sure without it I would agree that’s enough of a reason. Please provide evidence to support that this statement.

Thanks for your comment. The translation of languages has implications because translation of words may lose the nuances of what was said, and concepts in one language may be understood differently in another language. This is especially important for qualitative research which deals with textual data in all phases from data collection to analysis and publications. It is a challenge for all qualitative research, but when working with original data the researchers have the advantage of being able to recall the interview/focus group, are aware of the context and can check the words used in the original language. As systematic reviewers we are distant from the original data and would be unaware if our translation had subtly misinterpreted a quote or a theme. We now have provided a reference as you suggest. See the citation on page 10, line 175, reference # 43.

Reference 43: Van Nes F, Abma T, Jonsson H, Deeg D. Language differences in qualitative research: is meaning lost in translation? Eur J Ageing. 2010 Dec;7(4):313-6. [Page 26, line 542-543]

Q7. Line 181: It’s not actually stated who and how many people are conducting full-text review.

Two authors will independently screen all records, review reports and include studies for the review. We now have updated the sentence to provide clarity. The text reads: 

“Two authors (MNU and MH) will independently screen titles and abstracts, retrieve and review full-text papers for inclusion of studies against the eligibility criteria (see Table 2) using Covidence (www.covidence.org) [46]”. [Page 11, line 183-186]

Q8. Line 184 - 186: Doesn’t make sense, recommend a reword.

We have revised the sentence as you suggest which now reads: 

“Any disagreements that arise between the two reviewers (MNU and MH) at any stage of the study selection process will be resolved through discussion and involve the review team (HP, VH, KM, and JS) if necessary.” [Page 11, line 187-190]

Q9. Line 192: What is the reasoning for only having one author pilot the extraction form when two authors will be independently using it to extract the same data? I would advise having both authors pilot the form to make sure the same information is being extracted and ensure consistency.

Thanks for bringing this to our attention and for your suggestion. In fact, both review authors have worked jointly on developing and piloting the data extraction form, before extracting data from the included studies independently. We have updated the paragraph which now reads:

“Two review authors (MNU and MH) will pilot the data extraction form on at least one quantitative and one qualitative study before data are extracted from the remaining included studies using a refined form. We will extract data into a Microsoft Word and Excel file as necessary. Two review authors (MNU and MH) will independently extract data from all included studies (quantitative, qualitative and mixed-methods) and another author (HP/VH/KM/JS) will check accuracy of data transcribed into tables or meta-analyses. Any disagreement between MNU and MH relating to data extraction will be resolved by consensus. A third review author (HP/VH/KM/JS) will be involved to resolve any outstanding disagreement as necessary.” [Page 15, line 197-206]

Q10. Line 196: It states that two authors are conducting data extraction. It’s not clear if both authors are extracting all data or if they’re doing a proportion each that are then being checked by a third author, please clarify.

Thank you for highlighting the need for further clarity about data extraction. We have updated the paragraph under the ‘Data extraction and management’ section. For your convenience, you we reproduce the text below:

“Two review authors (MNU and MH) will independently extract data from all included studies (quantitative, qualitative and mixed-methods) and another author (HP/VH/KM/JS) will check accuracy of data transcribed into tables or meta-analyses. Any disagreement between MNU and MH relating to data extraction will be resolved by consensus. A third review author (HP/VH/KM/JS) will be involved to resolve any outstanding disagreement as necessary.” [Page 15, line 200-206]

Q11. Line 206: Are you collecting raw data (e.g.: means, standard deviation, mean difference). At the moment it’s unclear if you’re collecting summary data only.

Thanks for your comment. We now have updated the line 206 as you suggest. The text reads:

“Outcomes: primary and secondary outcomes specified and collected: e.g., means (standard deviation), median (interquartile range), confidence intervals, P-value, measurement scale used), and time points reported.” [Page 15, line 218-219]

Q12. Line 256: Risk of bias is discussed quite minimally, with no discussion of who is conducting it, how disagreements will be resolved, and how the information will be presented. Ideally there should be more information here.

Thanks for highlighting the need for additional information about the risk of bias assessment. At the beginning of the ‘Methodological quality assessment’ section (page 17, lines 245-248), we described “All included studies will be assessed for methodological quality independently and in duplicate by two review authors (MNU and MH). We will resolve any disagreements by discussion or by involving another author ((HP/VH/KM/JS).” We did not mention this information in the later subsections to avoid repetition. However, we now have updated the paragraph to increase clarity. The text now reads:

“All included studies (quantitative, qualitative and mixed-methods) will be assessed for methodological quality independently and in duplicate by two review authors (MNU and MH). Disagreement will be solved by discussion between the two authors (MNU and MH) or by involving another review author (HP/VH/KM/JS) if necessary.” [Page 17, lines 245-248]

Also, we have revised the subsection ‘Quantitative studies’ with additional information as you suggested. The paragraph reads:

“To assess the methodological quality of randomised controlled trials (RCTs), we will use the Cochrane risk of bias tool which assesses selection, performance, detection, attrition, reporting and other sources of bias enabling each study to be assigned as low; moderate; or high risk of bias [51]. We will record and tabulate a summary of the assessment with the overall judgement. For non-randomised studies we will use the Downs and Black checklist [52]. To reflect the relative weight of the quantitative findings, we will adopt the previously published approach of summarising three attributes for each study (design, population size, quality score) [53].” [Page 17, line 250-260]

Q13. Line 270: It’s stated that it’s not anticipated a meta-analysis will be possible, but as you don’t know that yet what is your plan if a meta-analysis is possible?

We agree with your comment. We now have the updated the paragraph which reads: 

“Based on our initial scoping we anticipate that our included studies will have substantial clinical, methodological and statistical heterogeneity and meta-analysis may not be appropriate. If that is the case, we will do a narrative synthesis to show the major outcomes and their relationships [51], illustrating findings graphically if appropriate [57]. However, if we find sufficient number of RCTs, we will perform meta-analysis for the clinical outcomes (asthma control, acute attacks, and asthma-related quality of life). One review author (MNU) will conduct the meta-analysis using Review Manager software (RevMan 2020, V.5.4.1) and another review author (MH) will check data accuracy. We will conduct a pooled quantitative synthesis for homogeneous data from RCTs using an inverse variance method and a random-effects model in the meta-analysis. If the included RCTs use the same outcome measurement tool, we will use pooled mean differences. However, if (as expected) outcome measurement tool varies among trials, we will consider standardised mean differences.” [Page 18-19, line 290-302]

Q14. Line 276: It is stated that thematic synthesis will be used without any information of what that practically means. Are you using line-by-line, who will be coding, are you going so far as to produce analytical themes or purely descriptive ones?

Thanks for highlighting the need for additional information about the qualitative data analysis. We now have updated the paragraph and the text reads:

“We will use thematic synthesis to combine the findings of studies that describe the views and experiences of patients and healthcare professionals on asynchronous asthma reviews. Following recognised methodology [58], two review authors (MNU and MH) will begin by familiarising themselves with the data against the aims of the review and note recurrent themes across the studies. After that they will develop a coding framework in discussion with the review team (HP, VH, KM, and JS). MNU and MH will independently perform line-by-line initial coding of the findings of the included studies (defined as all the text/quotes under the heading of ‘results’ or ‘findings’) translating the concepts from one study to another. They will then search for themes according to the predetermined thematic framework adding additional themes as they emerge. Analysis will be iterative and involve the multi-disciplinary author team before finalising the overarching themes and sub-themes. We will initially analyse patients and healthcare professionals’ data separately to identify, for example, conflicting views or experiences. If we find that the views and experiences are similar, we may combine the two subgroups in subsequent syntheses. We will generate tables of author-reported categories, themes, and subthemes regarding asynchronous online asthma reviews.” [Page 19-20, line 304-322]

Q15. Line 288: Double check your citations meet the PLOS ONE guidelines, the ‘Harden and colleagues [60]’ does not look correct.

Thanks for spotting this. We now have revised the sentence which reads:

“We will choose the appropriate methods and tools for integration as the review progresses following the Cochrane Qualitative and Implementation Methods Group guidance [60].” [Page 20, line 327-328]

Q16. In line with PLOS ONE’s data policy, please describe how the data will be made available at the point of publication.

Thank you for your comment. This is a systematic review protocol; no datasets were generated or analysed during the submission of the manuscript. A systematic review, by definition, presents data that are already published, but once the study is complete, we will publish our findings in a peer-reviewed journal and make all our summary tables and syntheses available. We now have added a ‘Data availability’ section in the manuscript and the text reads: 

“No datasets were generated or analysed for this systematic review protocol. Summary tables and syntheses from the review will be made publicly available in a peer-reviewed publication.” [Page 23, line 405- 408]

Reviewer #2: Amanda Ross-White

Q1. Thank you for this clearly written protocol with all necessary questions addressed. Helpful, but not needed, to identify who will be doing what at each stage, as this is only done for some tasks (eg. screening).

Thank you very much. We now have made this clear throughout the manuscript that two review authors (MNU and MH) will independently and in duplicate screen titles and abstract, review full-text papers, extract data, assess methodological quality of included studies, synthesise data, and assess confidence in evidence. Disagreements at any stage of the review will be resolved by discussion between the two authors (MNU and MH) or by involving another review author (HP/VH/KM/JS) as necessary. 

The revised text is reproduced in response to Qs 5, 7, 8, 9, 10, 12, 13, and14 above, and can be found in the revised manuscript on page 10: line: 164-166, page 11: line 183-190, page 15: line 197-206, page 17: line 245-248, page 19: line 296-298, 308-312.

Journal requirements.

1. We have checked that our formatting meets PLOS ONE's style requirements

2. We have amended the Data Availability statement (See response to Reviewer 1, Q16) 

3. We have checked our funding statements and corrected the discrepancy. The statement now reads: “The authors received no specific funding for this work. MNU is supported by a University of Edinburgh PhD Studentship (Grant number 34678) funded by the University of Edinburgh College of Medicine and Veterinary Medicine (CMVM) within the Asthma UK Centre for Applied Research (AUKCAR). The PhD studentship is nested in the IMP2ART (IMPlementing IMProved Asthma self-management as RouTine) programme at the University of Edinburgh (https://www.ed.ac.uk/usher/imp2art). The funders had no role in study design, data collection and analysis, decision to publish, or preparation of the manuscript.” [Page 22-23, line 387-395]

4. In response to reviewers’ comments, we have made following changes in the references:

• Ref #15: corrected in response to Reviewer 1: Q3

• Ref #43: added in response to Reviewer 1: Q6

• Ref #51: updated the reference #56 and cited as 51

---

## [Decision Letter · Decision Letter 1]

26 Jan 2023

Asynchronous digital health interventions for reviewing asthma: a mixed-methods systematic review protocol

PONE-D-22-22457R1

Dear Dr. Pinnock,

We’re pleased to inform you that your manuscript has been judged scientifically suitable for publication and will be formally accepted for publication once it meets all outstanding technical requirements.

Kind regards,

Heather Leggett

Academic Editor

PLOS ONE

Additional Editor Comments (optional):

Reviewers' comments:

Reviewer's Responses to Questions

**Comments to the Author**

1. Does the manuscript provide a valid rationale for the proposed study, with clearly identified and justified research questions?

Reviewer #1: Yes

Reviewer #2: Yes

2. Is the protocol technically sound and planned in a manner that will lead to a meaningful outcome and allow testing the stated hypotheses?

Reviewer #1: Yes

Reviewer #2: Yes

3. Is the methodology feasible and described in sufficient detail to allow the work to be replicable?

Reviewer #1: Yes

Reviewer #2: Yes

4. Have the authors described where all data underlying the findings will be made available when the study is complete?

Reviewer #1: Yes

Reviewer #2: Yes

5. Is the manuscript presented in an intelligible fashion and written in standard English?

Reviewer #1: Yes

Reviewer #2: Yes

6. Review Comments to the Author

You may also provide optional suggestions and comments to authors that they might find helpful in planning their study.

Reviewer #1: Thank you for addressing my previous comments, I am happy they have all been addressed sufficiently.

Reviewer #2: Thank you for the necessary changes. This paper is now ready for publication. Best wishes for 2023 added so I can meet the minimum character count :)

7. PLOS authors have the option to publish the peer review history of their article (what does this mean?). If published, this will include your full peer review and any attached files.

Reviewer #1: No

Reviewer #2: **Yes: **Amanda Ross-White

---

## [Editor Report · Acceptance letter]

31 Jan 2023

PONE-D-22-22457R1 

Asynchronous digital health interventions for reviewing asthma: a mixed-methods systematic review protocol 

Dear Dr. Pinnock:

I'm pleased to inform you that your manuscript has been deemed suitable for publication in PLOS ONE. Congratulations! Your manuscript is now with our production department. 

Kind regards, 

on behalf of

Dr. Heather Leggett 

Academic Editor

PLOS ONE